# Germline Polymorphisms Associated with Overall Survival in Lung Adenocarcinoma: Genome-Wide Analysis

**DOI:** 10.3390/cancers16193264

**Published:** 2024-09-25

**Authors:** Francesca Minnai, Sara Noci, Martina Esposito, Marc A. Schneider, Sonja Kobinger, Martin Eichhorn, Hauke Winter, Hans Hoffmann, Mark Kriegsmann, Matteo A. Incarbone, Giovanni Mattioni, Davide Tosi, Thomas Muley, Tommaso A. Dragani, Francesca Colombo

**Affiliations:** 1Institute for Biomedical Technologies, National Research Council, Segrate, 20054 Milan, Italyfrancesca.colombo@itb.cnr.it (F.C.); 2Department of Medical Biotechnology and Translational Medicine (BioMeTra), Università degli Studi di Milano, 20122 Milan, Italy; 3Department of Experimental Oncology, Fondazione IRCCS Istituto Nazionale dei Tumori, 20133 Milan, Italy; 4Translational Research Unit (STF), Thoraxklinik, University Hospital Heidelberg, 69120 Heidelberg, Germany; 5Translational Lung Research Center (TLRC), German Center for Lung Research (DZL), 69120 Heidelberg, Germany; 6Department of Thoracic Surgery, Thoraxklinik, University Hospital Heidelberg, 69120 Heidelberg, Germany; 7Department of Thoracic Surgery, Klinikum Rechts der Isar, Technische Universität München, 80333 Munich, Germany; 8Institute of Pathology, University Hospital Heidelberg, 69120 Heidelberg, Germany; 9Department of Surgery, Ospedale San Giuseppe, IRCCS Multimedica, 20099 Milan, Italy; 10Thoracic Surgery and Lung Transplantation Unit, Foundation IRCCS Ca’ Granda Ospedale Maggiore Policlinico, 20122 Milan, Italy

**Keywords:** GWAS, SNPs, Cox, eQTL, *NT5DC2*, *NAGK*

## Abstract

**Simple Summary:**

Our study aimed to understand why some people with lung adenocarcinoma, a type of lung cancer, survive longer than others, even when their conditions seem similar. By studying the DNA of 1464 patients, we found six specific genetic differences that appear to affect survival. These differences are located in parts of the DNA that do not directly make proteins but might control how certain genes are turned on or off. The goal is to use this information to predict which patients might have better outcomes and to develop more personalized treatment options. Further research is needed to confirm these results and to understand the underlying biological mechanisms, but our findings could eventually improve how we treat lung cancer and understand its outcomes better.

**Abstract:**

Background/Objectives: Lung cancer remains a global health concern, with substantial variation in patient survival. Despite advances in detection and treatment, the genetic basis for the divergent outcomes is not understood. We investigated germline polymorphisms that modulate overall survival in 1464 surgically resected lung adenocarcinoma patients. Methods: A multivariable Cox proportional hazard model was used to assess the association of more than seven million polymorphisms with overall survival at the 60-month follow-up, considering age, sex, pathological stage, decade of surgery and principal components as covariates. Genes in which variants were identified were studied in silico to investigate functional roles. Results: Six germline variants passed the genome-wide significance threshold. These single nucleotide polymorphisms were mapped to non-coding (intronic) regions on chromosomes 2, 3, and 5. The minor alleles of rs13000315, rs151212827, and rs190923216 (chr. 2, 3 and 5, respectively) were found to be independent negative prognostic factors. All six variants have been reported to regulate the expression of nine genes, seven of which are protein-coding, in different tissues. Survival-associated variants on chromosomes 2 and 3 were already reported to regulate the expression of *NT5DC2* and *NAGK*, with high expression associated with the minor alleles. High *NT5DC2* and *NAGK* expression in lung adenocarcinoma tissue was already shown to correlate with poor overall survival. Conclusions: This study highlights a potential regulatory role of the identified polymorphisms in influencing outcome and suggests a mechanistic link between these variants, gene expression regulation, and lung adenocarcinoma prognosis. Validation and functional studies are warranted to elucidate the mechanisms underlying these associations.

## 1. Introduction

Lung cancer is the leading cause of cancer death worldwide [1]. Nonetheless, lung cancer survival has progressively improved, with the 3-year survival rate increasing from 22% in cases diagnosed in 2004–2006 to 33% for patients diagnosed in 2016–2018 [2]. The histotype with the greatest improvement in survival (from 25% to 38%) is lung adenocarcinoma. Gains in survival are mainly due to earlier detection [3], advanced surgical procedures [4], better staging [5] and, particularly for non-small-cell lung cancer, the advent of targeted therapy and immunotherapy [6]. Despite these improvements, there is large variability in overall survival, for unknown reasons. The variability in outcomes is even observed among patients with the same tumor histotype and stage, and it is already known that age, sex and pathological stage are independent prognostic factors [7]. However, they do not fully explain the individual variability in prognosis due to the multifaceted nature of the disease [8].

It has been hypothesized that other prognostic factors affect survival, including a patient’s genetic background. One possible explanation for the different outcomes is that individual germline polymorphisms modulate still unknown genetic mechanisms affecting cancer growth and metastasis. To identify such polymorphisms, two analytical approaches can be used. In the candidate-gene approach, knowledge of pathological mechanisms prompts the search for polymorphisms in genes whose products are believed to be involved in cancer survival. This approach has already identified polymorphisms in genes that influence overall survival of lung cancer patients [9,10,11,12,13,14,15,16], but a replication of the findings is often lacking. The other approach to investigate the role of genetics in lung cancer survival uses unsupervised, genome- or exome-wide methods. So far, two studies of this type have identified polymorphisms and low-frequency variants associated with survival, although at a low statistical significance level [17,18]. 

Since the survival of lung cancer patients is a complex phenotype, the genome-wide approach is preferable to the candidate-gene approach, because it allows for the exploration of many variants in almost all genes and also in non-coding regions of the genome. The exome-wide approach is good for studying rare, low-frequency variants, but it does not provide information on non-coding regulatory variants. Genome- and exome-wide analyses require a large sample to achieve sufficient statistical power. As a result, studies that investigate survival in a homogeneous group of patients (e.g., those with the same histotype and genetic background) are often limited by an inadequate sample size. Thus, to obtain statistically robust results, we combined two European case series for a large genome-wide association study (GWAS) of 1464 lung adenocarcinoma patients, and explored, using a Cox model, the association of 7,265,396 imputed germline polymorphisms with overall survival at 60 months.

## 2. Materials and Methods

### 2.1. Case Series and Research Ethics

The study investigated two cases series of surgically resected lung adenocarcinoma patients from hospitals in the area around Milan, Italy, and in Heidelberg, Germany. Patients in Italy were enrolled between 1992 and 2022 at the Fondazione IRCCS Istituto Nazionale dei Tumori, San Giuseppe Hospital, and Fondazione IRCCS Cà Granda Ospedale Maggiore Policlinico. Patients in Germany were recruited at the Thoraxklinik between 2006 and 2015. These case series are a subset of those analyzed in our previous study on lung adenocarcinoma prognostic factors in 3078 patients [19]. 

Patients provided written informed consent to the use of their biological samples and data for research purposes, according to the European General Data Protection Regulation. The study was conducted in accordance with the Declaration of Helsinki and approved by the ethics committees of Fondazione IRCCS Istituto Nazionale dei Tumori (INT 224-17, on 19 December 2017), Ospedale San Giuseppe, IRCCS Multimedica (346.2018, on 1 October 2018), Fondazione IRCCS Ca’ Granda Ospedale Maggiore Policlinico (202_2019bis; on 12 March 2019) and University of Heidelberg (S-270/2001).

### 2.2. Clinical Data and Biological Samples

Clinical data were collected from all patients about sex, age at lung resection for adenocarcinoma, year of surgery, smoking habit, survival status 60 months after surgery, and pathological stage, based on the 6th to 8th editions of TNM staging criteria for lung cancer [20,21,22]. More in detail, the German patients were staged (or re-staged, for patients who had surgery before 2009) using criteria of the 7th edition, while the Italian patients were staged according to the edition that was valid at the moment of surgery. Smoking habit was reported as either “never smoker” or “ever smoker” (current or former smoker), because information on smoking cessation was not available for many patients.

Genomic DNA samples from patients recruited in Italy were already available at Fondazione IRCCS Istituto Nazionale dei Tumori, Milan, Italy, as they had been prepared from non-involved lung tissue using a DNeasy Blood and Tissue kit (Qiagen (Venlo, The Netherlands)), as previously described [17]. This DNA was fluorimetrically quantified with the Quant-iT PicoGreen dsDNA assay kit on an M1000 multiplate reader (Tecan (Zürich, Switzerland)). For patients in Germany, buffy coats were available at the Thoraxklinik biobank and used to extract genomic DNA using a FlexiGene DNA kit (Qiagen); this DNA was quantified using a Nanodrop ND-1000 spectrophotometer.

### 2.3. Genome-Wide Genotyping

Genome-wide genotype data for the entire sample of 1592 patients were collected separately on three subgroups and then merged for this study. Data were already available for 582 of the patients recruited in Italy, at Fondazione IRCCS Istituto Nazionale dei Tumori, Milan, Italy. These data had been obtained with Infinium Omni2.5-8 BeadChip microarrays (Illumina (San Diego, CA, USA)) on an Illumina HIscan System and using Illumina’s BeadStudio software v3, as described [23,24]. DNA from the remaining 530 patients from Italy and the 480 patients from Germany was genotyped using Axiom Precision Medicine Research Arrays (PMRA; Thermo Fisher Scientific) on GeneTitan instruments at two genotyping service providers, i.e., Thermo Fisher Scientific (Santa Clara, CA, USA) and the Functional Genomics facility of the Instituto de Investigaciones Biomédicas August Pi i Sunyer (IDIBAPS, Barcelona, Spain), for Italian and German samples, respectively. Axiom Analysis Suite software v5.2 (Thermo Fisher Scientific) was used to call genotypes on these samples using the “best practice” workflow (except for the average call rate threshold ≥  97).

Genotype data for the three subgroups were separately subject to preliminary genotype quality control (QC) using PLINK v.1.9 software [25] (Appendix A). For per-sample QC, we used an identity-by-descent test (as described in [26]) to identify and exclude related patients and duplicates. We also excluded samples with a call rate <98%, with sex discrepancies, or with excess heterozygosity (heterozygosity rate outside the range +/− 0.20). In per-marker QC, we removed variants with a genotyping call rate <98%, a minor allele frequency (MAF) < 1%, or a Hardy–Weinberg equilibrium test *p* < 1.0 × 10^−6^. 

Genotype imputation to whole-genome sequence (for autosomal variants) was carried out separately for the three subgroups using the Minimac4 algorithm on the TOPMed Imputation Server. Data were phased with Eagle v.2.4 software, GRCh38/hg38 was set as the array build, and TOPMed-r2 was set as the reference panel [27,28,29,30]. Genotypes imputed with an R^2^ ≤ 0.3 were considered of low-quality imputation [31] and thus filtered out together with those that had a MAF < 0.01. Finally, the three datasets were merged, retaining only biallelic variants with an imputed genotyping rate <98%.

PLINK 2 software [32] was used to perform a principal components analysis (PCA) on the entire dataset. The first 10 principal components (PCs) were used as covariates in survival analyses. The first four PCs were compared with those of 2504 samples from five populations (Africans, Americans, South-East Asians, East Asians, and Europeans) in the 1000 Genomes Project [33]. 

### 2.4. Statistical Analyses

The Italian and German series were compared using the chi-squared test (categorical variables) and Kolmogorov–Smirnov test (quantitative variables). Survival analyses were carried out in univariable and multivariable Cox proportional hazard models [34], using the coxph function of the survival package [35] in R environment. The following variables were considered: age (as both a quantitative and categorical variable), sex, pathological stage, country of enrollment, decade of surgery, smoking habit, and genotyping array. Data were censored at 60 months of follow-up. 

The GWAS survival analyses tested, in an additive model, the association between variants and patients’ overall survival limited at 60 months of follow-up. The GenABEL package in R environment [36] was used to test the multivariable Cox proportional hazard model with genotypes, using the first 10 PCs, sex, age, decade of surgery and pathological stage as covariates. The genome-wide statistical threshold was set at *p* < 5.0 × 10^−8^. A suggestive threshold was considered at *p* < 1.0 × 10^−5^. The Benjamani–Hochberg method of false discovery rate (FDR) [37] was used to correct for multiple testing. 

Kaplan–Meier curves were plotted using the survfit function of the same survival package, and the log-rank test was used to assess significance. These analyses were performed using genotypes coded as in a dominant model. A two-sided *p*-value < 0.05 was set as the statistical significance threshold for these analyses.

A multivariable Cox proportional hazard model with the significant clinical variables and the top-significant SNPs was also tested. A backward stepwise model selection, based on the Akaike information criterion (AIC), was performed using the stepAIC function of MASS package, in R, to identify independent prognostic factors. The significance threshold for this analysis was *p* < 0.05. 

### 2.5. In Silico Functional Analyses

The identified germline polymorphisms (associated with overall survival at *p* < 1.0 × 10^−5^) were investigated for a possible regulatory role, by searching for them in two public expression quantitative-trait locus (eQTL) databases (both accessed on 5 February 2024): GTEx (Analysis V8 release, GTEx_Analysis_v8_eQTL_EUR.tar) and eQTLGen [38] (https://www.eqtlgen.org/cis-eqtls.html). 

Genes reported as being regulated by the variants that were in this case found to associate with survival (at *p* < 5.0 × 10^−8^) were selected for further analysis as possible prognostic factors. The Kaplan–Meier Plotter [39] online tool was used to look for associations between these genes’ expression levels and survival according to published gene expression data from lung adenocarcinoma tumor tissue (accessed on 12 February 2024). The following parameters (different from default settings) were used: follow-up threshold of 60 months, adenocarcinoma histology, and multivariable Cox regression with stage and sex as covariates. With these settings, the analyses were performed using data from 534 lung adenocarcinoma patients. High and low expression groups were defined by dichotomizing at the median value of log_2_-transformed probe intensities. A two-sided *p* < 0.005 was set as the significance threshold for this analysis.

## 3. Results

Genomic DNA from 1592 surgically resected lung adenocarcinoma patients was genotyped, but 113 samples were excluded in the QC steps (Appendix A). Thus, data for 1479 patients were used in survival analyses. This cohort included 1049 patients from Italy (71%) and 430 from Germany (29%) (Table 1). The patients had a median age at surgery of 65 years, and there was a slight abundance of males (62.4%). Only 15.3% had never smoked, while the remaining 80.4% were classified as ever-smokers (current or former smokers). More than half of patients (52.6%) had an early-stage tumor (pathological stage I). Regarding the period in which lung resection had been carried out, less than 15% of cases were surgically operated between 1992 and 2000, 37.8% were treated between 2001 and 2010, and 47.2% underwent surgery after 2010. The median follow-up period was 54 months, and by 60 months only 62.7% were still alive.

The subgroups of patients enrolled in Italy and Germany were different (Table 1). Patients from Italy were older than those from Germany (median, 66 vs. 63 years; Kolmogorov–Smirnov test, *p* < 0.001) and more likely to be male (64.7% vs. 56.5%, chi-squared test, *p* = 0.004). This latter observation may be due to the selection period (later for the German series): indeed, lung cancer in women has been increasing in recent years [40]. The proportions of never- and ever-smokers were similar between the groups. Most patients enrolled in Italy (58.7%) had a stage I tumor, while most of those in the German series had tumors of stages II-IV (61.4%, chi-squared *p* < 0.001). Regarding the period of surgery, the German series did not comprise patients operated before 2006, but there was no significant difference in the proportion of patients enrolled in the 2000s or after 2010 between the two series (chi-squared *p* = 0.82). The median follow-up period was shorter for patients in Italy than Germany (50 vs. 60 months; Kolmogorov–Smirnov test *p* = 0.011); this difference was mainly due to the incomplete 60-month follow-up for the most recently enrolled patients from Italy. A greater percentage of patients from Italy were alive at the 60-month follow-up (65.3% vs. 56.5%; chi-squared test *p* = 0.002); this might be due to the longer median follow-up period or the higher percentage of more advanced stages in the German series. Finally, all patients from Germany were genotyped using the Axiom PMRA array, while the genotyping of patients from Italy was carried out with either the Infinium Omni2.5-8 or Axiom PRMA array. 

To identify factors that affect the survival of lung adenocarcinoma patients, we first did a survival analysis using univariable proportional hazard Cox regression (Table 2). Sex, pathological stage, and the decade of surgery were the most significant prognostic factors. Survival probability also depended on the type of genotyping array, with a lower risk of death for cases genotyped on the Infinium Omni2.5-8 array. This difference may be attributed to the fact that approximately two-thirds of patients genotyped with these arrays had stage I tumors (Table 1). To identify independent prognostic factors, we also did multivariable proportional hazard Cox regression. Due to missing data for 75 patients, this model was run on 1404 patients. In this analysis, age, sex, pathological stage and the decade of surgery were independent prognostic factors, whereas the country of enrollment and genotyping array did not associate with survival. In detail, the mortality risk increased with age (hazard ratio [HR] = 1.02, *p* < 0.001). The prognostic effect of age was more evident when this variable was treated as categorical: indeed, patients in the age groups 65–74 and ≥75 had higher mortality risks than younger patients (HR = 1.33 and HR = 1.82, respectively). Prognosis was better for females than males (HR = 0.66, *p* < 0.001). Increasing pathological stage was associated with the highest mortality risk, with 2-, 4-, and about 6-fold higher HRs for stage II, III and IV tumors, respectively, than patients with stage I tumors (*p* < 0.001). Overall survival was longer for patients who underwent resection more recently, with a consequential drop in HR from 1.0 for patients treated before 2000 to 0.68 for those treated between 2001 and 2010 and 0.48 for those treated after 2010 (*p* < 0.001).

Based on these analyses, in the GWAS, Cox regression with genotypes was carried out using age, sex, pathological stage, and the decade of surgery as covariates, and the first 10 PCs to correct for population stratification. A plot of the first four PCs of our series, along with those of 2504 samples from five populations (Africans, Americans, South-East Asians, East Asians, and Europeans) in the 1000 Genomes Project [33], is given in Appendix A, to visualize which ancestral group our patients belonged to. 

### 3.1. Germline Variants Associated with Overall Survival 

The genotype data of patients from the Italian and German series were used in the genome-wide survival analysis. After preliminary QC (Appendix A) and data imputation, the whole dataset comprised information on 7,265,396 germline polymorphisms and 1464 patients (15 patients were excluded due to missing data regarding pathological stage or decade of surgery). Each variant was independently tested in an additive multivariable Cox model, and 224 single nucleotide polymorphisms (SNPs) were found to associate with overall survival at *p* < 1.0 × 10^−5^ (Figure 1, Appendix A). Among them, six SNPs, on chromosomes 2, 3 and 5, passed the genome-wide statistical significance threshold (*p* < 5.0 × 10^−8^). These SNPs are (in order of increasing *p*-value): rs74464684 (HR = 2.8), rs13000315 (HR = 2.5), rs71414848 (HR = 2.5), rs76553845 (HR = 2.7), rs151212827 (HR = 2.6), and rs190923216 (HR = 2.9). Because these SNPs have HRs greater than 1, their minor alleles are negative prognostic factors. An increasing number of their minor alleles are associated with an at least 2-fold higher risk of death (HR > 2). 

Since the MAF of these six SNPs was <5% (Appendix A), we grouped heterozygous patients with patients who were homozygous for the minor allele and drew Kaplan–Meier survival curves (Figure 2). Patients carrying at least one minor allele of each SNP had worse prognosis than patients homozygous for the major allele (log-rank test, *p* < 0.001). Thus, these low-frequency variants have a dominant effect on the risk of death from lung adenocarcinoma.

Five of the six top-ranking SNPs (excluding rs190923216 on chromosome 5) map near other variants associated with survival, although at a lower significance level (*p* < 1.0 × 10^−5^). Indeed, the analysis identified 19 variants on chromosome 2 in a region <30 kbp (Figure 3A). The top-ranking variant, rs13000315, was in linkage disequilibrium (LD) with all the other variants (r^2^ > 0.6 and D′ > 0.7) except rs56354394. On chromosome 3, the analysis identified 279 variants in a region <600 kbp (Figure 3B), and the top-ranking variant, rs74464684, was in LD with the other variants (r^2^ > 0.5 and D′ > 0.7).

Finally, we tested, in a stepwise multivariable Cox model, the association of survival with age, sex, stage, decade of surgery and genotype of the six top-ranking SNPs, to understand if they were independent prognostic factors. The four clinical variables were all independent prognostic factors. Of the six top-ranking SNPs, three (rs13000315, rs151212827, and rs190923216) were independently associated with overall survival (Table 3).

### 3.2. SNPs Associated with Lung Adenocarcinoma Survival Have Regulatory Roles

The 224 significant SNPs mapped in non-coding regions of the genome (mostly intronic; Appendix A), so we hypothesized that they participate in the regulation of gene expression. Therefore, we searched in two eQTL databases and found that 73 and 128 of them were already identified as eQTLs in GTEx and eQTLGen, respectively. 

According to GTEx, the 73 SNPs regulate the expression of 34 mRNAs in 48 tissues, for a total of 1125 eQTLs (Appendix A). Limiting the analysis to lung tissue, 16 SNPs (one on chromosome 5, six on chromosome 7, and nine on chromosome 10) had been reported as lung eQTLs of four genes: the minor allele of the SNP on chromosome 5 (rs190923216) was associated with higher expression levels of two mRNAs (the antisense RNA, *CKMT2-AS1*, and the pseudogene *RPS12*), the six SNPs on chromosome 7 affect the expression of *COPG2*, and the nine SNPs on chromosome 10 influence the expression of a long non-coding RNA, *LINC00865*. 

According to eQTLGen, the 128 SNPs regulate the expression of 43 unique mRNAs in blood, for a total of 543 eQTLs (Appendix A). Of note, the top five ranking SNPs from our genome-wide analysis (on chromosomes 2 and 3) had already been reported in eQTLGen as regulating the mRNA levels of seven coding genes (Table 4). In this database, the minor alleles of rs13000315 and rs71414848 on chromosome 2 correlate positively with the levels of *CLEC4F*, *NAGK*, *MCEE*, and *CD207*. Moreover, the three SNPs on chromosome 3 (rs74464684, rs76553845, and rs151212827) associate with the expression of *NT5DC2*, *TKT*, and *UQCC5*: increasing numbers of the minor allele of these three SNPs correlate positively with the expression of *NT5DC2* and negatively with the expression of *TKT* and *UQCC5*.

The combined results from GTEx and eQTLGen databases identified a total of 11 genes, including eight coding genes that were regulated by the six top-ranking SNPs in our genome-wide survival analysis. Analyzing the expression levels of these coding genes in the tumor tissue of lung adenocarcinoma patients (in publicly available databases), we observed that four were associated with overall survival. High expression levels of *NT5DC2*, *TKT*, *UQCC5*, and *NAGK* genes were negative prognostic factors (multivariable Cox and log-rank *p* < 0.005; Appendix A). Thus, there was an agreement between the direction of effect of gene expression on prognosis and that of the minor allele on gene expression for *NAGK* and *NT5DC2*, whereas this was not the case for *TKT* and *UQCC5*. 

## 4. Discussion

This study explored the association between germline polymorphisms and survival, 60 months after surgery for lung adenocarcinoma, in a series of 1464 patients. A multivariable Cox proportional hazard model identified six SNPs whose genotype associated with overall survival at the genome-wide significance threshold and whose minor allele was a negative prognostic factor (HR > 1). Three of these SNPs (rs13000315, rs151212827, rs190923216) were independent prognostic factors, together with the patients’ age, sex, pathological stage, and decade of surgery. All six top-ranking SNPs had already been reported as regulators of gene expression. 

Before doing the genome-wide survival analysis, we examined the patients’ clinical data and confirmed that age, sex and pathological stage were independent prognostic factors for lung adenocarcinoma [7]. In addition, as expected for a series of patients who were enrolled over a large time interval, the probability of survival was associated with the decade in which they had surgery. In our analysis, smoking habit did not affect survival, in contrast to previous reports, e.g., [41,42]. However, because we did not have complete data about the patients’ smoking history (e.g., pack years and smoking cessation data for former smokers), we may have underestimated the effect of smoking on survival. 

The study identified six SNPs associated with survival at the genome-wide significance level. Because they map to non-coding regions of the genome, we hypothesized that they have regulatory roles in gene expression. Indeed, five SNPs were previously reported as eQTLs targeting coding genes: the two variants on chromosome 2 associate with the expression levels of *CLEC4F*, *NAGK*, *MCEE*, and *CD207* genes, and the three variants on chromosome 3 are eQTLs of *NT5DC2*, *TKT*, and *UQCC5* genes. The expression levels of these genes correlate directly with the number of minor alleles of the regulatory SNPs, except for *TKT* and *UQCC5*, which correlate inversely. 

It has already been demonstrated that tumor expression of *NT5DC2* is a prognostic marker of lung adenocarcinoma, with high levels of expression associated with poor survival [43]. It has also been observed that this gene has a role in non-small-cell lung cancer progression: indeed, its overexpression promoted the proliferative, migratory, and invasive capacities of NSCLC cells, whereas its down-regulation induced cell cycle arrest and apoptosis [44]. We found three SNPs (rs74464684, rs76553845, and rs151212827) whose minor allele associated with worse prognosis. The data already published indicated that these minor alleles were associated with high levels of *NT5DC2*, and that, in tumor tissue, high levels of *NT5DC2* were associated with poor prognosis. These associations suggest that individuals with a minor allele of these germline polymorphisms are more susceptible to a negative lung adenocarcinoma outcome than patients homozygous for the major allele, possibly due to a higher expression of this gene, which might promote tumor cell proliferation, migration and invasion. The minor alleles of these same SNPs, associated with poor survival, have also been reported to negatively regulate the expression of *TKT* and *UQCC5* genes. However, high expression levels of these genes in lung adenocarcinoma have been associated with poor overall survival: *TKT* has been suggested to be a negative prognostic marker in lung adenocarcinoma [45], and *UQCC5* (alias *SMIM4*) has been proposed, together with another six polymorphisms, as a survival prediction tool [46]. Therefore, it is more plausible that the SNPs on chromosome 3 affect lung adenocarcinoma survival through the regulation of expression of *NT5DC2* gene.

In the literature, we did not find evidence of a prognostic role of the four genes (*CLEC4F*, *NAGK*, *MCEE*, and *CD207*) regulated by the top-significant SNPs on chromosome 2 (rs13000315 and rs71414848). Nonetheless, we found that, in lung adenocarcinoma tissue, *NAGK* (N-acetylglucosamine kinase) expression levels are associated with overall survival. Indeed, patients expressing high levels (above the median) of *NAGK* had a higher death probability than those expressing low levels. As our patients with at least one copy of the minor allele of rs13000315 and rs71414848 (associated with higher levels of *NAGK*) had a negative lung adenocarcinoma outcome, we speculate that this was due to a genetic predisposition to higher *NAGK* expression than in patients homozygous for the major alleles of these SNPs. *NAGK* is a metabolic enzyme involved in the salvage of hexosamine, and, in particular, of N-acetylglucosamine (GlcNAc), a precursor of UDP-GlcNAc that is needed by cells to modify many proteins and, thus, work properly. Tumor cells have a high requirement for this metabolite due to their proliferation. In a cellular model of cancer different from lung (i.e., pancreatic), knocking down *NAGK* limited tumor growth, since the absence of this gene did not allow for a quick recycling of UDP-GlcNAc [47]. It would be interesting to investigate this functional role of *NAGK* in lung tumor cells, to validate our findings that patients with high expression levels of this gene (due to germline regulatory variants that genetically predispose to higher *NAGK* expression) had poor prognosis. 

A limitation of our study is the lack of information about other possible prognostic factors, such as the somatic mutational status and therapies administered to the patients in addition to surgical resection. Unfortunately, these data were missing for a rather large number of patients, so we preferred not to reduce the sample size of our GWAS. Nevertheless, we believe that our results are promising.

Functional studies are needed to test the hypothesized biological mechanism of action of the identified SNPs in modulating survival. It would be interesting to test whether the already reported eQTLs act in lung adenocarcinoma tissue. It would also be useful to understand, for instance, the role of rs190923216 (on chromosome 5), which has been reported to be an eQTL for an anti-sense gene (*CKMT2-AS1*) in normal lung. Studies are needed to understand whether this gene plays a role in lung adenocarcinoma prognosis. Finally, validation in an independent, but homogeneous series is needed; to achieve this, it will be important to consider that our findings were obtained from the analysis of prevalently European patients and that the allele frequencies of the identified variants were quite low.

## 5. Conclusions

Our study identified germline variants affecting lung adenocarcinoma patient survival, possibly due to a regulatory role on gene expression, in particular of *NT5DC2* and *NAGK* genes. Indeed, their expression in lung adenocarcinoma tissue was previously reported to associate with poor prognosis. Overall, our results underscore the significant role of genetic factors in predisposing lung adenocarcinoma patients to different outcomes.

## Figures and Tables

**Figure 1 cancers-16-03264-f001:**
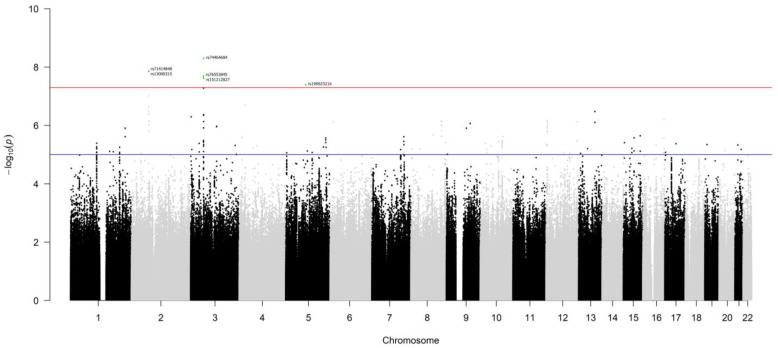
Manhattan plot of the results of the GWAS for overall survival, at 60 months of follow-up, of lung adenocarcinoma patients. Germline polymorphisms are plotted on the x-axis according to their genomic position (GChr 38, hg38 release) and on the y-axis according to their association with survival probability (−log_10_(*p*-values)). The horizontal red and blue lines represent the threshold of genome-wide significance (*p* < 5.0 × 10^−8^) and a suggestive threshold at *p* < 1.0 × 10^−5^, respectively. Genome-wide significant SNPs are highlighted in green and annotated with the rsID from dbSNP.

**Figure 2 cancers-16-03264-f002:**
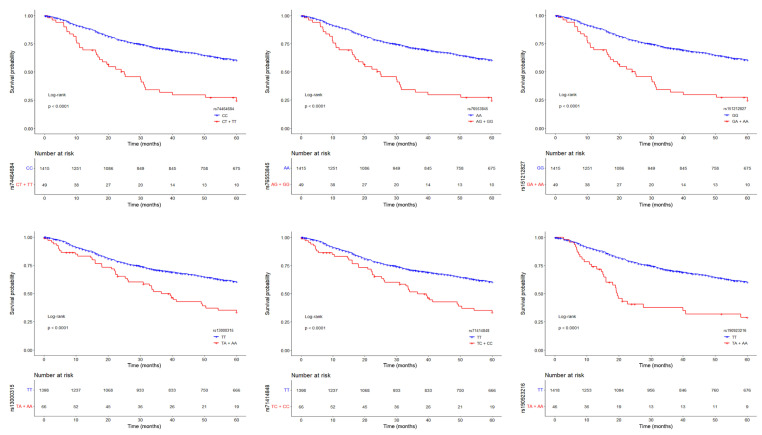
Kaplan–Meier survival curves (at 60 months of follow-up) for lung adenocarcinoma patients according to the genotype of the six top-significant variants (rs74464684, rs76553845, rs151212827, rs13000315, rs71414848, and rs190923216, from top left to bottom right), coded as in a dominant model. Red lines represent patients homozygous for the minor alleles and heterozygotes, while blue lines represent patients homozygous for the major alleles. Crosses denote censored samples. Below each plot are indicated the numbers of patients at risk in the genotype groups. Log–rank *p*-values are shown.

**Figure 3 cancers-16-03264-f003:**
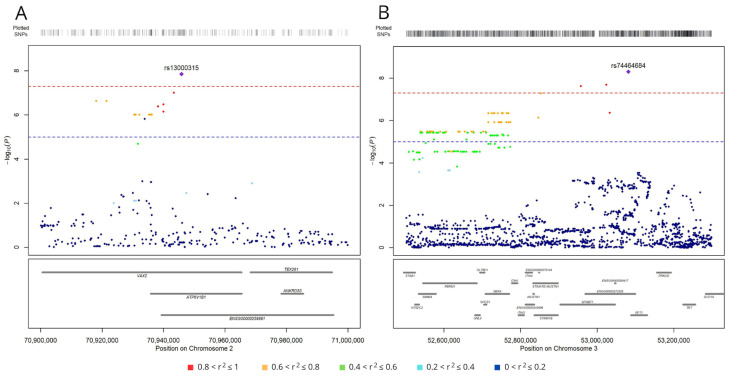
Regional association plots for SNPs associated with survival (at *p* < 1.0  ×  10^−5^) on chromosomes 2 (**A**) and 3 (**B**). SNPs are plotted on the x-axis according to their genomic position (GChr 38, hg38 release), and −log_10_[*p*-values] for their association with overall survival are plotted on the y-axis. Horizontal dashed red and blue lines represent the genome-wide threshold of significance (*p* < 5.0 × 10^−8^) and the suggestive threshold (*p* < 1.0 × 10^−5^), respectively. Dot color represents the level of linkage disequilibrium, expressed as r^2^, between each SNP and the top variants (purple diamond in panel A, rs13000315, and in panel B, rs74464684).

**Table 1 cancers-16-03264-t001:** Characteristics of patients with lung adenocarcinoma, by country of enrollment.

Characteristic			Total (*n* = 1479)	Italian Series (*n* = 1049)	German Series (*n* = 430)	*p*
Age at surgery, years, median (range)			65 (30–90)	66 (30–85)	63 (37–88)	<0.001 ^a^
Age group, years, *n* (%)						<0.001 ^b^
	<55		227 (15.3)	139 (13.3)	88 (20.5)	
	55–64		496 (33.5)	347 (33.1)	149 (34.7)	
	65–74		563 (38.1)	413 (39.3)	150 (34.9)	
	≥75		193 (13.0)	150 (14.3)	43 (10.0)	
Sex, *n* (%)						0.004 ^b^
	Male		922 (62.4)	679 (64.7)	243 (56.5)	
	Female		557 (37.6)	370 (35.3)	187 (43.5)	
Smoking habit, *n* (%)						0.67 ^b^
	Never		225 (15.3)	160 (15.3)	65 (15.1)	
	Ever		1190 (80.4)	826 (78.7)	364 (84.7)	
	Missing		64 (4.3)	63 (6.0)	1 (0.2)	
Pathological stage, *n* (%)						<0.001 ^b^
	I		778 (52.6)	612 (58.7)	166 (38.6)	
	II		257 (17.4)	174 (16.3)	83 (19.3)	
	III		348 (23.6)	195 (18.5)	153 (35.6)	
	IV		82 (5.5)	54 (5.2)	28 (6.5)	
	Missing		14 (0.95)	14 (1.3)	0 (0)	
Decade of surgery, *n* (%)						
	Before 2000		221 (14.9)	221 (21.1)	0 (0)	0.82 ^b^
	2001–2010		558 (37.8)	370 (35.3)	188 (43.7)	
	After 2010		699 (47.2)	458 (43.6)	241 (56.0)	
	Missing		1 (0.1)	0 (0)	1 (0.2)	
Follow-up, months, median (IQR)			54 (21–60)	50 (20–60)	60 (23–60)	0.011 ^a^
Survival status at 60 months, *n* (%)						0.002 ^b^
	Alive		928 (62.7)	685 (65.3)	243 (56.5)	
	Dead		551 (37.3)	364 (34.7)	187 (43.5)	
Genotyping array, *n* (%)						<0.001 ^b^
	Infinium Omni2.5-8					
		All tumors	559 (37.7)	559 (53.3)	0 (0)	
		Stage I tumors *	366 (65.5)	366 (65.5)	0 (0)	
	Axiom PMRA					
		All tumors	920 (62.3)	490 (46.7)	430 (100)	
		Stage I tumors *	412 (44.8)	246 (50.2)	166 (38.6)	

^a^ Kolmogorov–Smirnov test; ^b^ chi-squared test; IQR, interquartile range; PMRA, Precision Medicine Research Array; * Percentage calculated on the subset of samples genotyped with the Infinium Omni2.5-8 array or Axiom PMRA.

**Table 2 cancers-16-03264-t002:** Factors associated with overall survival, according to univariable and multivariable Cox regression.

Characteristic	Univariable Analyses ^a^		Multivariable Analyses ^b^	
	HR (95% CI)	Cox *P*	HR (95% CI)	Cox *P*
Age, years	1.00 (1.00–1.01)	0.40	1.02 (1.01–1.03)	2.93 × 10^−5^
Age group, years				
<55	1.00		1.00	
55–64	0.84 (0.66–1.08)	0.18	1.06 (0.82–1.37) *	0.62 *
65–74	0.91 (0.71–1.16)	0.45	1.33 (1.03–1.71) *	0.028 *
≥75	1.08 (0.80–1.46)	0.62	1.82 (1.31–2.52) *	2.98 × 10^−4^ *
Sex				
Male	1.00		1.00	
Female	0.61 (0.51–0.74)	1.88 × 10^−7^	0.66 (0.54–0.81)	6.53 × 10^−5^
Smoking habit				
Never	1.00		1.00	
Ever	1.25 (0.98–1.60)	0.078	1.05 (0.81–1.37)	0.72
Pathological stage				
I	1.00		1.00	
II	2.06 (1.61–2.64)	9.01 × 10^−9^	2.05 (1.60–2.65)	2.38 × 10^−8^
III	4.04 (3.30–4.94)	<2.00 × 10^−16^	4.14 (3.35–5.12)	<2.00 × 10^−16^
IV	5.95 (4.45–7.97)	<2.00 × 10^−16^	5.59 (4.12–7.57)	<2.00 × 10^−16^
Decade of surgery				
Before 2000	1.00		1.00	
2001–2010	0.77 (0.62–0.95)	0.017	0.68 (0.53–0.86)	0.0014
After 2010	0.45 (0.36–0.57)	2.5 × 10^−11^	0.48 (0.37–0.63)	8.62 × 10^−8^
Country				
Italy	0.85 (0.71–1.01)	0.063	0.96 (0.77–1.21)	0.73
Germany	1.00		1.00	
Genotyping array				
Infinium Omni2.5-8	0.79 (0.66–0.94)	0.0075	0.84 (0.68–1.04)	0.11
Axiom PRMA	1.00		1.00	

CI, confidential interval; HR, hazard ratio; ^a^ For 1479 patients; ^b^ For 1404 patients.; * Results from a multivariable model with age coded as a categorical variable. The summary statistics of this model, for the other listed variables, were very similar to those shown in the table, which instead referred to the multivariable model with age as quantitative variable.

**Table 3 cancers-16-03264-t003:** Hazard ratios for overall survival in 1464 lung adenocarcinoma patients (15 patients were excluded due to missing data).

Variable	HR (95% CI)	Cox *P*
Age	1.02 (1.01–1.03)	2.8 × 10^−6^
Sex		
Male	1.00	
Female	0.68 (0.56–0.82)	8.0 × 10^−5^
Pathological stage		
I	1.00	
II	2.00 (1.56–2.57)	4.8 × 10^−8^
III	4.34 (3.53–5.34)	<2.0 × 10^−16^
IV	6.24 (4.62–8.41)	<2.0 × 10^−16^
Decade of surgery		
Before 2000	1.00	
2001–2010	0.70 (0.56–0.88)	2.4 × 10^−3^
After 2010	0.49 (0.39–0.63)	2.4 × 10^−8^
Genomic variant ^a^		
rs13000315	2.62 (1.92–3.56)	9.6 × 10^−10^
rs151212827	2.32 (1.63–3.29)	2.8 × 10^−6^
rs190923216	2.58 (1.75–3.79)	1.5 × 10^−6^

HR, hazard ratio; CI, confidential interval; ^a^ SNPs were entered under an additive model, with the minor allele being the effect allele.

**Table 4 cancers-16-03264-t004:** Evidence from eQTLGen that the top five ranking SNPs are *cis*-eQTLs in blood.

SNP	Chr.	Minor Allele	Major Allele	Regulated Gene	*p*-Value	Z-Score	FDR
rs13000315	2	A	T	CLEC4F	5.08 × 10^−53^	15.3	0
		A	T	NAGK	4.30 × 10^−10^	6.24	0
		A	T	MCEE	3.74 × 10^−6^	4.63	0.0100
		A	T	CD207	1.50 × 10^−5^	4.33	0.0387
rs71414848	2	C	T	CLEC4F	3.16 × 10^−53^	15.4	0
		C	T	NAGK	3.74 × 10^−10^	6.26	0
		C	T	MCEE	4.81 × 10^−6^	4.57	0.0130
		C	T	CD207	1.88 × 10^−5^	4.28	0.0470
rs74464684	3	T	C	NT5DC2	6.54 × 10^−49^	14.7	0
		T	C	TKT	8.97 × 10^−16^	−8.04	0
		T	C	UQCC5	4.80 × 10^−7^	−5.03	0.00147
rs76553845	3	G	A	NT5DC2	4.40 × 10^−45^	14.1	0
		G	A	TKT	2.18 × 10^−12^	−7.02	0
		G	A	UQCC5	4.12 × 10^−7^	−5.06	0.00122
rs151212827	3	A	G	NT5DC2	1.57 × 10^−47^	14.5	0
		A	G	TKT	1.08 × 10^−11^	−6.80	0
		A	G	UQCC5	2.76 × 10^−8^	−5.56	0.000121

Chr, chromosome; FDR, false discovery rate.

## Data Availability

The data presented in this study are available on request from the corresponding author due to privacy restrictions.

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
