# Peer review of "Germline Polymorphisms Associated with Overall Survival in Lung Adenocarcinoma: Genome-Wide Analysis"

_cancers, 2024, doi:10.3390/cancers16193264_

Round 1

Reviewer 1 Report

Comments and Suggestions for Authors

The article submitted for review is very interesting and brings new information to the research on the role of molecular changes in lung cancer. However, before the article is published, the authors should make minor changes:

- introduce access dates for in-silico analysis tools

- replace the wording sex with gender in the entire manuscript

- Figure 3 - do A and B refer to chromosomes or rs ?? because it is not entirely clear

- in the discussion section it would be useful to add information/characterization regarding the significant polymorphisms found and the genes that these found have an impact on; additionally, it would be useful to characterize the genes, especially NT5DC2 and NAGK, and explain in what mechanism they may participate in carcinogenesis and affect survival in lung cancer?

Author Response

Question 1 - introduce access dates for in-silico analysis tools

Answer 1. The dates of access to the online tools (i.e., eQTLGen, GTEx, and Kaplan-Meier Plotter) were already reported in the Methods section (page 4, lines 190 and 197).

Question 2 - replace the wording sex with gender in the entire manuscript

Answer 2. Our study did not collect information about patients’ gender but biologically determined sex. Therefore, we have not replaced the word “sex” with “gender”.

Question 3 - Figure 3 - do A and B refer to chromosomes or rs ?? because it is not entirely clear

Answer 3. Panels A and B refer to chromosomes 2 and 3, respectively, as already indicated on the x-axes and stated in the title of the figure legend. We edited the figure legend to clarify that the purple diamonds in the panels are SNPs (indicated by their rsID), and not “headings”.

Question 4 - in the discussion section it would be useful to add information/characterization regarding the significant polymorphisms found and the genes that these found have an impact on; additionally, it would be useful to characterize the genes, especially NT5DC2 and NAGK, and explain in what mechanism they may participate in carcinogenesis and affect survival in lung cancer?

Answer 4. In the revised manuscript, we discussed further what the reviewer suggested (page 12, lines 403-404; page 13, lines 415, 421-430).

Reviewer 2 Report

Comments and Suggestions for Authors

The authors presented a study identifying germline mutations associated with lung adenocarcinoma patient survival. They found six variants to be significant and independent negative prognostic factors. Overall, the manuscript appears intriguing and well-written. I have a few comments:

1. All figures in this paper are kind of blurred and hard to read. The authors should provide high-resolution and quality figures in this paper.

2. In Figure 1, it would be beneficial to annotate and highlight the six significant SNPs, to make it easier for readers capture the major findings.

Comments on the Quality of English Language

Minor editing of English language required.

Author Response

Question 1. All figures in this paper are kind of blurred and hard to read. The authors should provide high-resolution and quality figures in this paper.

Answer 1. We uploaded the figure files at high resolution.

  1. In Figure 1, it would be beneficial to annotate and highlight the six significant SNPs, to make it easier for readers capture the major findings.

Answer 2. As requested by the reviewer, we annotated and colored the six top variants, to highlight them (revised Figure 1).